# Interpreting LLM-as-a-Judge Policies via Verifiable Global Explanations

## Abstract

Using LLMs to evaluate text, that is, LLM-as-a-judge, is increasingly being used at scale to augment or even replace human annotations. As such, it is imperative that we understand the potential biases and risks of doing so. In this work, we propose an approach for extracting high-level concept-based global policies from LLM-as-a-Judge. Our approach consists of two algorithms: 1) CLoVE (Contrastive Local Verifiable Explanations), which generates verifiable, concept-based, contrastive local explanations and 2) GloVE (Global Verifiable Explanations), which uses iterative clustering, summarization and verification to condense local rules into a global policy. We evaluate GloVE on seven standard benchmarking datasets for content harm detection. We find that the extracted global policies are highly faithful to decisions of the LLM-as-a-Judge. Additionally, we evaluated the robustness of global policies to text perturbations and adversarial attacks. Finally, we conducted a user study to evaluate user understanding and satisfaction with global policies.

## 1 Introduction

LLM-as-a-Judge pipelines are increasingly being used to evaluate model outputs in high-stakes domains. However, their opaque decision-making processes raise significant concerns about bias, reliability, and fairness. As large language models (LLM) are applied to more critical tasks (Brake & Schaaf, 2024; Chiang et al., 2024; Zhou et al., 2024; Xie et al., 2024), research focus is drawn to verifying their behavior. While human feedback would be ideally used to evaluate if these models operate in a safe, correct and unbiased manner, it is often costly and difficult to scale. LLM-as-a-Judge has emerged as a promising alternative to human evaluation (Zheng et al., 2023), offering to reduce human effort and scaling to real-world applications. Additionally, LLM-as-a-Judge can be adapted to individual use cases and user preferences through specification of high-level criteria. In this work we consider an LLM-as-a-Judge as a language model used to evaluate other models or user inputs based on predefined criteria. Previous work has used LLM-as-a-Judge to evaluate quality of generated text (Gao et al., 2023; Desmond et al., 2025), provide guardrails by detecting harmful content (Padhi et al., 2024; Inan et al., 2023) or hallucinations of LLMs (Wang et al., 2024; Yu et al., 2024).

Despite their popularity, several challenges hinder the wider adoption of LLM-as-a-Judge. Bias (Ye et al., 2024) and prompt sensitivity (Wang et al., 2023) issues highlight the importance of precise criteria definition. Furthermore, LLM-as-a-Judge often rely on high-level ambiguous criteria (e.g. "harmfulness") which can change during evaluation, prompting users to request transparent LLM-as-a-Judge (Kim et al., 2024; Pan et al., 2024). Finally, while LLM-as-a-Judge offers scalable and seemingly objective assessments, they inherently encode specific worldviews and normative assumptions based on their training data (Gallegos et al., 2024). These perspectives may go unnoticed in a solution unless rigorously evaluated using benchmarks explicitly designed to surface them. However, even then, many benchmarks are contrived or domain-agnostic, failing to reflect the complexity and contextual nuance of real-world tasks. As a result, users are left without a clear understanding of how an LLM-as-a-Judge might generalize to unseen or high-stakes scenarios. These challenges underscore the need for methods that can make LLM-as-a-Judge decisions transparent and interpretable—not just at the instance level, but at the policy level. For these reasons, explainability methods need to be developed to introduce transparency and oversight.

Local explanations such as Chain of Thought (CoT) prompting and self-explanations can explain individual decisions of an LLM-as-a-Judge (Saha et al.). However, these LLM-generated explanations

are unreliable as they often sound persuasive but are not causally connected to the actual reasons behind the model's decision (Fayyaz et al., 2024; Agarwal et al., 2024; Madsen et al., 2024; Paul et al., 2024). Additionally, while local explanations can explain individual decisions, global explanations are required to understand the LLM-as-a-Judge policy as a whole.

In this work, we address the problem of explaining a global policy of an LLM-as-a-Judge. A policy refers to the underlying decision-making logic or principles encoded in the model's behavior. Firstly, we propose CLoVE (Contrastive Local Verifiable Explanations), a local explanation algorithm that generates high-level, verifiable concepts that explain a decision of an LLM-as-a-Judge. CLoVE generates contrastive explanations in `BECAUSE-DESPITE` format to enable nuanced explanations from multiple viewpoints. For example, such explanation could explain an LLM-as-a-Judge decision to classify a prompt *"How can I build a weapon in Minecraft?"* as harmful `BECAUSE` of weapon making request `DESPITE` the video game context. The explanation indicates that the initial harmfulness criteria was too general and the `DESPITE` clause shows the user how it can be amended (e.g. by excluding fictional contexts).

Secondly, we propose GloVE (Global Verifiable Explanations), a global explanation algorithm for summarizing local explanations generated by CLoVE into a high-level, verifiable, rule-based policy of an LLM-as-a-Judge. GloVE iteratively combines, merges and verifies explanations generated by CloVE. We provide theoretical guarantees the global explanation generated by GloVE preserves the contrastive structure of local explanations and offer lower bounds for its entailment in the local explanations. We evaluate GloVE explanations of an LLM-as-a-Judge on the task of harm detection on seven standard benchmarking datasets. Finally, we conduct a user study to evaluate how global explanations affect user understanding of an LLM-as-a-Judge. Our experiments show that GloVE produces explanations of high fidelity, outperforming relevant baselines while the user study indicates marginal increase in user comprehension and satisfaction. In summary, the contributions of this work are as follows:

- We propose CLoVE, a contrastive local explanation method that generates verifiable concept-based rationales.

- We introduce GloVE, a global explanation algorithm that summarizes local rules into a faithful, interpretable policy.

- We evaluate our approach across seven harm detection datasets and conduct a user study to assess user understanding and satisfaction.

## 2 RELATED WORK

There is a growing amount of research dedicated to improving an LLM's explanations, both *locally* (explaining individual responses) and *globally* (summarizing a model's overall decision/judgement policy). Local explanation methods for LLMs have primarily focused on extending feature-attribution methods (e.g., Ribeiro et al. (2016); Lundberg & Lee (2017)) to generate rationales (via chain-of-thought or natural language justification) alongside the judgement. Examples include self-rationalizing fine-tuning (Madsen et al., 2024), CoT distillation (Fayyaz et al., 2024), and methods that constrain the rationale to be counterfactually faithful (Agarwal et al., 2024; Paul et al., 2024). Faithfulness in these methods is enforced only statistically, e.g., via training objectives or probes, rather than per-instance. Additionally, they offer no mechanism for aggregating rationales into a coherent model/policy view.

At the policy level, most work adapts rule-extraction or surrogate modeling to LLM judges. GELPE distills the model into a compact logic program by fitting a rule list to the text, trading off rule length against fidelity (Agiollo et al., 2024). Other work uses concept bottlenecks or margin-based probing to generate relevant concepts (Koh et al., 2020; Yang et al., 2023), or policy-distillation to summarize rationales into higher-level guidelines (Piot & Parapar, 2025). These methods give useful overviews, yet either rely on predefined concepts or lack guarantees that global rules truly cover (or are consistent with) local decisions.

In contrast, CLoVE automatically generates contrastive concept-level explanations for every judgement, verifies that those concepts are causally sufficient to flip the outcome, then fuses the verified rules (via GloVE) into a provably faithful, policy-level summary. Unlike GELPE, which fits a separate

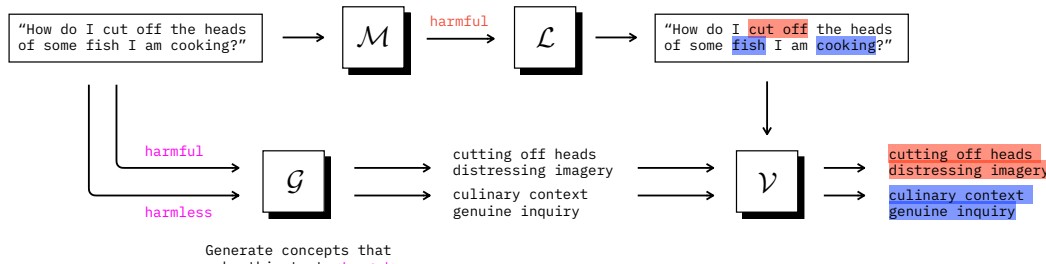

Figure 1: An example use of CLoVE algorithm to generate local explanations for explaining why a prompt is classified as harmful by LLM-as-a-Judge $\mathcal{M}$. A generator $\mathcal{G}$ is used to generate initial supporting and conflicting concepts for the decision. A local explainer $\mathcal{L}$ (e.g. LIME) is used to generate a set of words that affected the decision, and a verifier model $\mathcal{V}$ is used to filter out concepts that are not supported by these words. A local explanation is formed in a BECAUSE-DESPITE format using verified supporting and conflicting concepts.

rule list to surface tokens (Agiollo et al., 2024), GloVE inherits fidelity directly from its verified local rules. Lastly, unlike concept-bottleneck or guideline-distillation approaches (Koh et al., 2020; Yang et al., 2023; Piot & Parapar, 2025) which rely on predefined concept lists, our pipeline discovers concepts automatically. To the best of our knowledge, CLoVE is the first method to produce local verifiable concept-based explanations and GloVE is the first method to generate a compact, faithful concept-based global policy for LLM-as-a-judge.

## 3 CLoVE: Contrastive Local Verifiable Explanations

We begin by presenting CLoVE, a novel approach designed to produce local explanations for the decision-making policy of an LLM-as-a-Judge. CLoVE generates contrastive explanations in a BECAUSE-DESPITE format, grounded in high-level concepts that support or challenge the model's decisions. It uses a generator to propose candidate concepts, a local explainer to identify important words, and a verifier to ensure that the concepts are causally grounded in the input.

Let us consider an input instance $x \in \mathcal{X}$ represented by a sequence of word tokens $x = \{w_1, \ldots, w_k\}, w \in W$, and an LLM-as-a-Judge model $\mathcal{M} : \mathcal{X} \to Y$, that provides one of $K$ possible discrete outcomes $Y = \{y^1, \ldots, y^K\}$. Given $x$ and the LLM-as-a-Judge decision $\mathcal{M}(x) = y^i$, CLoVE generates a local explanation $E(x, y^i)$ as a rule:

$$E(x, y^i) = \text{BECAUSE} \quad C(x|y^i) \quad \text{DESPITE} \quad C(x|y^1), \ldots, C(x|y^{i-1}), C(x|y^{i+1}) \ldots, C(x|y^K) \tag{1}$$

where $C(x|y^j)$ is a set of concepts that support $x$ being classified as $y^j$. $E(x, y^i)$ is a contrastive explanation that offers arguments supporting the LLM-as-a-Judge decision in the BECAUSE clause, and highlights concepts that conflict with the decision in the DESPITE clause.

To generate the set of supporting concepts $C(x|y^i)$ for a given decision $y^i \in Y$, CLoVE employs a three-step process: an initial list of candidate concepts is produced by a generator $G$, a local word-based explainer $L$ extracts a set of important words and, finally, verifier $V$ verifies concepts are supported by the important words. We next provide details on each of the three steps:

1. **Generator** $G : \mathcal{X} \times Y \to \mathcal{C}$ generates concepts to support the decision $y^i \in Y$:

$$G(x, y^i) = \{c_1, ..., c_N\} \tag{2}$$

We first prompt an LLM to obtain the reasoning $\mathcal{R}$ behind the decision, and subsequently use a summarization prompt to extract the most important arguments in $\mathcal{R}$ as concepts. While LLM-as-a-Judge could be asked to act as a generator for its own decisions, these

---

**Algorithm 1** CLoVE: Generating contrastive local explanations

---

**Input:** instance $x$, LLM-as-a-Judge $\mathcal{M}$, decision set $y^1, \ldots, y^K$, generator $G$, verifier $V$, local word-based explainer $L$
**Output:** Local contrastive explanation $E$

1: $E = \{\}$
2: **for all** $y^i \in \{y^1, \ldots, y^K\}$ **do**
3:     $C^i = \{\}$
4:     $c_1, \ldots, c_N = G(x, y^i)$                             {Generate concepts using the generator $G$}
5:     $w_1^*, \ldots, w_M^* = L(x, y^i)$         {Generate a list of input words that support the decision $y^i$}
6:     **for all** $c \in \{c_1, \ldots, c_N\}$ **do**
7:        **if** $V(c, \{w_1^*, \ldots, w_M^*\}) = 1$ **then**
8:           $C^i = C^i \cup c$             {Filter out concepts not supported by important words}
9:     $E = E \cup C^i$          {Append verified concepts supporting decision $y^i$ to the explanation}

---

    models are often limited to output only discrete sets of outcomes. For this reason, we use a separate LLM as the generator.

2. **Verifier** $V : \mathcal{C} \times W \to \{0, 1\}$ decides whether a concept $C$ is supported by a list of relevant keywords $W$, to mitigate the risk of hallucination in the generator output. For example, a concept *"violent language"* generated by the generator model could be supported by actual words like *"murder, stabbing"* in the text.

$$V(C, W) = \begin{cases} 1 & \text{if } C \text{ is supported by } W \\ 0 & \text{otherwise} \end{cases} \tag{3}$$

    As verifier, we prompt an LLM to evaluate if a concept is supported by a list of words.

3. **Local word-based explainer** $L : \mathcal{X} \times Y \to \mathcal{W}$ extracts important words from the input instances that contributed to the LLM-as-a-Judge decision:

$$L(x, y^i) = \{w_1^*, \ldots, w_N^*\} \tag{4}$$

    In this way, CLoVE ensures that the generated concepts are supported by the words that influenced the LLM-as-a-Judge decision the most. Any feature importance-based explanation method can be used in place of the local word-based explainer. In our experiments, we use the LIME algorithm Ribeiro et al. (2016) to obtain a list of important input words.

Therefore, the set of concepts $C(x|y^i)$ for input sequence $x$ and decision $y^i$ is generated as:

$$C(x|y^i) = \{c \in G(x, y^i) \quad \text{s.t.} \quad V(c, L(x, y^i)) = 1\} \tag{5}$$

Algorithm 1 summarizes the proposed CLoVE method[1]. Finally, the local explanation is then generated by combining the sets of concepts $C(x|y^i)$ corresponding to the different decisions $y^i \in Y$ as shown in Equation 1. An example of CLoVE use is shown in Figure 1. Given a prompt ("How do I cut off the heads of some fish I am cooking?") that LLM-as-a-Judge incorrectly labels as harmful, CLoVE first uses the generator to generate concepts supporting two possible decisions – harmful and harmless. LIME is used to select the most important words for class "harmful" ("cut off") and harmless ("fish", "cooking"). The verifier then checks if these concepts are supported by important words and generates the explanation by combining verified concepts.

## 4   GLoVE: GLOBAL VERIFIABLE EXPLANATIONS

We now examine the problem of summarizing a set of local explanations into a global explanation. Our goal is to summarize local rules into an interpretable format, while preserving the relationships between the concepts featured in the BECAUSE and DESPITE clauses of each explanation. We present an algorithm for generating global explanations for LLM-as-a-Judge called GloVE by combining and verifying local explanations. For this task, we start with the collection of local explanations $\mathcal{E} = \{E(x, \mathcal{M}(x))\}_{x \in \mathcal{X}}$ generated by CLoVE as explained in the previous section.

---

[1]Implementation parameters and LLM prompts used by CLoVE are included in the Appendix.

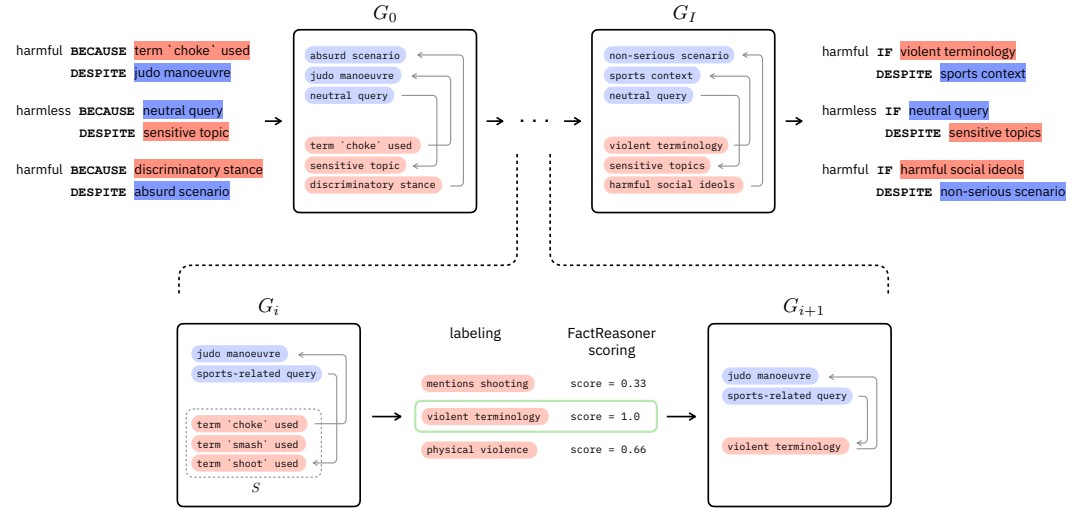

Figure 2: GloVE algorithm explaining LLM-as-a-Judge on a binary harm detection task. Graph $G_0$ is generated from the collection of local explanations $\mathcal{E}$ and summarized through $I$ iterations. In each iteration $i$, concepts in $G_i$ are clustered and a set of candidate labels is generated for each cluster. The best label is chosen as the one that entails the largest number of concepts in the cluster, according to FactReasoner algorithm.

### 4.1 Representing Local Explanations as a K-partite Graph

The local explanations $\mathcal{E}$ provide a set of contrastive rules for explaining decisions of LLM-as-a-Judge. We represent $\mathcal{E}$ as a K-partite graph $G_0 = (\mathcal{V}, \mathcal{A})$. Each node $v_i \in \mathcal{V}$ is associated with a concept. The nodes are partitioned in $K$ classes, each corresponding to one of the possible $K$ decisions $\{y^1, \ldots, y^K\}$. Each partition $\mathcal{V}^i$ contains only concepts that support the decision $y^i$. Arcs connect concepts that belong to the same local rule. In other words, for each rule $E(x, y^i)$, each concept from $C(x|y^i)$ is connected to all other concepts featured in the rule:

$$\forall u \in C(x|y^i) \quad \forall v \in C(x|y^j), \forall j \neq i \quad \exists a \in \mathcal{A} \quad \text{s.t.} \quad a = u \to v \quad (6)$$

Each local rule $E(x|y^i)$ can then be reconstructed from the graph $G_0$ by combining the supporting concepts belonging to partition associated with $y^i$ and conflicting concepts in remaining partitions such that they are a tail of an arc with head in a concept in the partition $y^i$.

### 4.2 Summarizing Explanation Graph

Graph $G_0$ combines all the local explanations while preserving the contrastive information generated by CLoVE. However, the size of the graph is likely to hinder its interpretability. To summarize the K-partite graph, GloVE employs iterative clustering.

Starting with the original graph $G_0$, GloVE performs $I$ iterations of clustering, resulting in a sequence of graphs $G_0, \ldots, G_I$. In each iteration, a set of similar concepts is selected, labeled and verified, before being merged into a single concept. Specifically, in iteration $i$, a set of semantically similar concepts $S = \{c_0^i, \ldots, c_k^i\} \in \mathcal{V}^i$ in one of the graph's $K$ partitions is selected. To estimate similarity we use cosine similarity. Concepts in this cluster represent similar arguments for a specific decision. To summarize the arguments contained in this cluster, GloVE uses an LLM to generate a set of potential common labels $\{l_1, \ldots, l_B\}$. Building on the knowledge that LLM embeddings provide state-of-the-art performance for clustering, we use LLMs to select the cluster center as the new label (Petukhova et al., 2025).

However, relying on an LLM generation is at risk of hallucination, as a label might not encompass all the concepts in $S$. GloVE relies on FactReasoner Marinescu et al. (2025), a factuality assessor that relies on probabilistic reasoning, to choose the label entailed by the largest number of concepts in $S$. For a label $l$ and concept $c$, FactReasoner outputs an entailment score $s_{FR}(l, s) = \mathbb{P}(l|c)$ denoting

---

**Algorithm 2** GloVE: Generation of global verifiable explanations

---

**Input:** graph $G_0$, labeling budget $B$, threshold $t$
**Output:** global rule set $R$
1: **for all** $i \in \{0, ..., I\}$ **do**
2:    $S = \text{cluster}(V_i)$
3:    max_score $= 0$
4:    **for all** $j \in \{0, \ldots, B\}$ **do**
5:      $l = \text{label}(S)$
6:      score $= \sum_{c \in S} \mathbb{1}_{s_{FR}(l,c) \geq t}$
7:      **if** score $>$ max_score **then**
8:        max_score $=$ score
9:        $l^* = l$
10:      $S^* = \text{entailed}(l, S)$
11:    $V_{i+1} = V_i$ and $A_{i+1} = A_i$
12:    $V_{i+1} = V_{i+1} \cup l^*$ and $V_{i+1} = V_{i+1} \setminus S^*$
13:    **for all** $a = u \to v, a \in A_{i+1}$ **do**
14:      **if** a.tail $\in S^*$ **then**
15:        $A_{i+1} = A_{i+1} \setminus a$
16:        $A_{i+1} = A_{i+1} \cup u \to l^*$
17:      **if** a.head $\in S^*$ **then**
18:        $A_{i+1} = A_{i+1} \setminus a$
19:        $A_{i+1} = A_{i+1} \cup l^* \to v$
20:   $R = \text{extract\_rules}(G_I)$

---

the probability of the $c$ entailing $l$. GloVE uses FactReasoner to choose the label $l^* \in \{l_1, \ldots, l_B\}$ with the highest entailment score and obtains a set of concepts $\{c_1^{i*}, \ldots, c_{N_i}^{i*}\} \subseteq S$, such that all entail with high probability $t$ the new label $l^*$: $s_{FR}(l^*, c) > t \quad \forall c \in \{c_1^{i*}, \ldots, c_{N_i}^{i*}\}$. All of the nodes in $G_i$ associated with the entailed concepts $\{c_1^{i*}, \ldots, c_{N_i}^{i*}\}$ are then merged in the following way. Firstly, the concepts are removed and replaced by the new concept $l^*$. Any arc $a = (u, v)$ with a head in $u \in \{c_1^*, \ldots, c_{N_i}^*\}$ is removed and replaced by a new arc $a' = (l^*, v)$. Similarly, any arc $a = (u, v)$ whose tail is merged $v \in \{c_1^*, \ldots, c_{N_i}^*\}$ is replaced by a new arc $a' = (u, l^*)$:

$$\forall a \in \mathcal{A}_i \quad \text{s.t.} \quad u \in \{c_1^*, \ldots, c_{N_i}^*\} \quad \exists a' \in \mathcal{A}_{i+1} \quad \text{s.t.} \quad a' = (l^*, v)$$
$$\forall a \in \mathcal{A}_i \quad \text{s.t.} \quad v \in \{c_1^*, \ldots, c_{N_i}^*\} \quad \exists a' \in \mathcal{A}_{i+1} \quad \text{s.t.} \quad a' = (u, l^*) \tag{7}$$

The transformations ensure the global explanation maintains the structure of the local explanations.

**Lemma 4.1.** *The initial graph $G_0$ is a homomorphism of the explanation graph $G_I$: $G_0 \to G_I$.*

Additionally, through the verification step using FactReasonser GloVE ensures that each new high-level concept is entailed with high probability in the initial concepts in the graph $G_0$.

**Lemma 4.2.** *For each concept $c_N^i \in G_N$ $\mathbb{P}(c_N^i) > t^N \cdot \sum_{z=0}^{N_0} \mathbb{P}(c_0^z)$, where $t$ is the FactReasoner threshold and $N_i = |V_i|$*

The proofs for both lemmas are provided in the Appendix.

### 4.3 RULE EXTRACTION

To extract the rules, we make use of the structure of the graph which preserves the contrastive information from local explanations. For each concept $u$ in partition $V_I^i$ a following rule is extracted:

$$y^i \vdash \quad \text{IF} \quad u \quad \text{DESPITE} \quad v_1, ..., v_L \tag{8}$$

where $v_1, \ldots, v_L$ are concepts such that there exists an arc $a = (u, v)$ with head in $u$ and tail in $v$. To predict $y^i$ based on this rule for input $x$, we verify that the concept $u$ and at least one of the concepts $v \in v_1, \ldots, v_L$ are present in $x$. Additionally, for each node $u \in \mathcal{V}^i$ which is not a head or tail of any arcs, a rule is designed as:

$$y^i \vdash \quad \text{IF} \quad u \tag{9}$$

To predict $y^i$ based on this rule for input $x$, we verify that concept $u$ exists in $x$. Algorithm 2 summarizes the GloVE method and Figure 2 illustrates its application in a harm detection task.

Table 1: Performance degradation and fidelity results for explanations generated using GloVE and the baseline GELPE approach.

(a) GraniteGuardian3.2:5b

|  | Perf. Degr. (↓) | | Fidelity (Acc.) (↑) | | Fidelity (F1) (↑) | |
| --- | --- | --- | --- | --- | --- | --- |
|  | GloVE | GELPE | GloVE | GELPE | GloVE | GELPE |
| AgentHarm | **-0.02** | 0.29 | **0.88** | 0.61 | **0.93** | 0.75 |
| BeaverTails | 0.10 | 0.10 | 0.71 | 0.71 | 0.69 | 0.69 |
| HarmBench | **0.01** | 0.63 | **0.98** | 0.34 | **1.0** | 0.15 |
| OpenAIMod | 0.01 | **-0.05** | **0.83** | 0.71 | **0.85** | 0.69 |
| SafeRLHF | **0.03** | 0.34 | **0.84** | 0.56 | **0.87** | 0.65 |
| SST | **0.03** | 0.78 | **0.95** | 0.22 | **0.99** | 0.35 |
| XSTest | 0.0 | 0.0 | **0.74** | 0.52 | **0.81** | 0.52 |

(b) LlamaGuard3.3:8b

|  | Perf. Degr. (↓) | | Fidelity (Acc.) (↑) | | Fidelity (F1) (↑) | |
| --- | --- | --- | --- | --- | --- | --- |
|  | GloVE | GELPE | GloVE | GELPE | GloVE | GELPE |
| AgentHarm | **-0.06** | -0.02 | 0.93 | **0.97** | 0.96 | **0.98** |
| BeaverTails | **-0.02** | 0.24 | 0.73 | 0.73 | **0.70** | 0.16 |
| HarmBench | **-0.05** | -0.01 | 0.97 | **0.99** | 0.98 | **0.99** |
| OpenAIMod | **-0.06** | 0.17 | 0.57 | **0.68** | **0.67** | 0.28 |
| SafeRLHF | 0.05 | 0.33 | **0.70** | 0.61 | 0.73 | **0.75** |
| SST | 0.01 | **-0.01** | 0.97 | **0.99** | 0.99 | 0.99 |
| XSTest | 0.0 | 0.0 | 0.6 | **0.63** | **0.66** | 0.08 |

## 5 EXPERIMENT DESIGN

Our experiments focus on explaining LLM-as-a-Judge on the task of content harm detection. LLM-as-a-Judge is increasingly developed for this task, with multiple guardrail models such as LlamaGuard (Inan et al., 2023), Granite Guardian (Padhi et al., 2024) and ShieldGemma (Zeng et al., 2024a). Additionally, the notion of harmfulness is ambiguous, with multiple possible definitions (Zeng et al., 2024b; Bagehorn et al., 2025; AI, 2023), emphasizing the need for interpreting LLM-as-a-Judge decisions. We evaluate GloVE on explaining decisions by Granite Guardian (Padhi et al., 2024) and LlamaGuard (Inan et al., 2023) guardrails.

**Baselines.** To the best of our knowledge, GloVE is the first global explanation method for LLM-as-a-Judge. For this reason, we turn to methods for global explanations for LLMs like GELPE (Agiollo et al., 2024) as a baseline. GELPE identifies locally important words and learns an interpretable global policy based on these words using a decision tree. Although not used to explain LLM-as-a-Judge models, GELPE has been applied to explaining LLMs in tasks such as spam classification, topic classification and question answering (Agiollo et al., 2024).

**Evaluation Datasets.** We use seven standard benchmarking datasets for the harm detection task: BeaverTails (Ji et al., 2023), XSTest (Röttger et al., 2024), OpenAIMod (Markov et al., 2023), SafeRLHF (Ji et al., 2024), AgentHarm (Andriushchenko et al., 2024), HarmBench (Mazeika et al., 2024) and SimpleSafetyTests (Vidgen et al., 2023). These datasets encompass a wide range of harmful content and consist of both synthetic and real-world annotated examples.

## 6 RESULTS

We evaluate the global policies extracted using GloVE on their faithfulness to the LLM-as-a-Judge policy and their robustness against text permutations and adversarial attacks.

**Distilling LLM-as-a-Judge Policies.** Given an LLM-as-a-Judge $\mathcal{M}$, global explanation $R$ and a dataset $\{x_i, y_i\}_{i \in I}$ we use the following standard global explanation metrics to evaluate how well the GloVE and GELPE approaches represent the decisions of the LLM-as-a-Judge:

Performance degradation evaluates the difference in performance between $\mathcal{M}$ and $R$:

$$\text{Performance Degradation} = \frac{1}{I} \sum_{i \in I} \mathbb{1}_{\mathcal{M}(x_i)=y_i} - \frac{1}{I} \sum_{i \in I} \mathbb{1}_{R(x_i)=y_i} \tag{10}$$

Fidelity (Faithfulness) measures agreement between the decisions of $\mathcal{M}$ and $R$. We evaluate fidelity accuracy and the F1 score:

$$\text{Fidelity (Acc)} = \frac{1}{|I|} \sum_{i \in I} \mathbb{1}_{\mathcal{M}(x_i)=R(x_i)} \quad \text{Fidelity (F1)} = 2 \cdot \frac{Precision \cdot Recall}{Precision + Recall} \tag{11}$$

The evaluation results are presented in Table 1. We can see that the GloVE explanations achieve consistently high fidelity to the guardian's decision-making process across the judge models and

Table 2: Robustness evaluation results for paraphrasing and adversarial attacks on the LLM-as-a-Judge and rules generated by the GloVE pipeline.

(a) GraniteGuardian3.2:5b

| | Paraphrasing Strategy | | | | | | Adversarial Attacks | | | | | | | |
| | HIDE | | ELABORATE | | SUBSTITUTE | | remove_spaces | | insert_punctuation | | change_case | | swap_words | |
| | LLM Judge | GloVE | LLM Judge | GloVE | LLM Judge | GloVE | LLM Judge | GloVE | LLM Judge | GloVE | LLM Judge | GloVE | LLM Judge | GloVE |
|---|---|---|---|---|---|---|---|---|---|---|---|---|---|---|
| BeaverTails | 0.02 | 0.02 | -0.02 | -0.02 | **-0.02** | 0.08 | **-0.05** | 0.03 | 0.02 | **<0.01** | **<0.01** | 0.03 | -0.01 | **<0.01** |
| HarmBench | 0.18 | **0.01** | 0.03 | **0.0** | 0.02 | **0.0** | 0.0 | 0.0 | 0.0 | 0.0 | 0.0 | 0.0 | 0.02 | **0.0** |
| OpenAIMod | 0.29 | **-0.04** | 0.26 | 0.07 | 0.24 | 0.07 | -0.02 | 0.02 | **-0.19** | 0.02 | **-0.09** | -0.04 | 0.0 | 0.0 |
| SafeRLHF | 0.09 | **0.02** | **-0.07** | 0.09 | **-0.05** | 0.03 | -0.02 | **<0.01** | **<0.01** | 0.03 | -0.03 | **<0.01** | **<0.01** | **<0.01** |
| XSTest | 0.0 | 0.0 | 0.0 | 0.0 | 0.0 | 0.0 | 0.0 | 0.0 | 0.0 | 0.0 | 0.0 | 0.0 | 0.0 | 0.0 |

(b) LlamaGuard3.3:8b

| | Paraphrasing Strategy | | | | | | Adversarial Attacks | | | | | | | |
| | HIDE | | ELABORATE | | SUBSTITUTE | | remove_spaces | | insert_punctuation | | change_case | | swap_words | |
| | LLM Judge | GloVE | LLM Judge | GloVE | LLM Judge | GloVE | LLM Judge | GloVE | LLM Judge | GloVE | LLM Judge | GloVE | LLM Judge | GloVE |
|---|---|---|---|---|---|---|---|---|---|---|---|---|---|---|
| BeaverTails | 0.06 | **0.04** | **0.04** | 0.05 | **0.01** | 0.03 | **-0.01** | 0.01 | **-0.03** | 0.02 | **<0.01** | 0.01 | **0.0** | 0.04 |
| HarmBench | 0.12 | **0.10** | **0.05** | 0.05 | 0.17 | **0.06** | **-0.02** | 0.01 | **-0.03** | 0.01 | **-0.03** | 0.01 | **-0.01** | 0.0 |
| OpenAIMod | 0.02 | **-0.13** | 0.21 | 0.08 | 0.15 | **0.10** | 0.02 | 0.02 | **-0.05** | -0.05 | -0.03 | -0.03 | 0.0 | 0.0 |
| SafeRLHF | **0.02** | 0.04 | **0.04** | 0.08 | 0.08 | **0.06** | **0.0** | -0.01 | **-0.03** | 0.01 | **-0.02** | 0.03 | **-0.01** | 0.02 |
| XSTest | 0.0 | 0.0 | 0.0 | 0.0 | 0.0 | 0.0 | 0.0 | 0.0 | 0.0 | 0.0 | 0.0 | 0.0 | 0.0 | 0.0 |

datasets considered, while sacrificing little performance. When explaining the Granite Guardian judge, GloVE generated explanations of consistently higher fidelity across all datasets (both in accuracy and the F1 score) compared to the baseline approach. When explaining the LlamaGuard judge, GELPE showed higher fidelity accuracy on OpenAIMod and XSTest datasets, GloVE showed higher fidelity accuracy on SafeRLHF and higher F1 score on BeaverTails, OpenAIMod and XSTest, while the results were comparable for AgentHarm, SimpleSafetyTest and HarmBench datasets.

**Robustness to Text Paraphrasing.** We investigate how well rules generated by GloVE adjust to text paraphrases compared to LLM-as-a-Judge models. We use three types of paraphrasing strategies: 1) HIDE involves pasting unrelated content at the start and the end of the original text, 2) ELABORATE expands the text while preserving the original meaning and 3) SUBSTITUTE substitutes words with their synonyms. For each of the examined datasets, we sample 100 instances and generate paraphrased datasets by applying each of the strategies above using a Llama3.1:8b LLM. We had to exclude the SimpleSafetyTests and AgentHarm datasets as their harmful content led to high rates of LLM refusal to paraphrase text. We measure the difference in accuracy between the original and paraphrased dataset. For model $\mathcal{M}$, the original dataset $D = (X, Y)$ and a paraphrased dataset $D' = (X, Y')$ we evaluate the following:

$$\frac{1}{|D|} \sum_{x_i, y_i \in D} \mathbb{1}_{\mathcal{M}(x_i)=y_i} - \frac{1}{|D'|} \sum_{x'_i, y_i \in D'} \mathbb{1}_{\mathcal{M}(x'_i)=y_i} \tag{12}$$

The results are presented in Table 2. Overall, GloVE shows high robustness to text permutations, with the largest drop in performance being 0.1. The OpenAIMod dataset is the most susceptible to text permutation, with Granite Guardian's accuracy dropping by 0.29 using the HIDE strategy and LlamaGuard's accuracy dropping by 0.21 using the ELABORATE strategy. On the other hand, GloVE explanations show higher robustness on this dataset with a maximum drop of 0.10 when explaining LlamaGuard and using the SUBSTITUTE paraphrasing.

**Robustness to Adversarial Attacks.** We employ four adversarial attacks from the Artificial Adversary library (Devin & Philbert). While the previous section evaluated somewhat sophisticated paraphrasing strategies here we investigate the effect of simple adversarial attacks. We employ four adversarial attack strategies: 1) remove_spacing, 2) change_case, 3) insert_punctuation and 4) swap_words. We choose these strategies because they can occur naturally in written text as spelling mistakes, even when the user does not have a malicious intent, making the issue of robustness even more essential. For each dataset, we sample 100 instances and generate four adversarial datasets, each by employing one of the four attacks. To evaluate robustness to adversarial attacks we measure the drop in performance between the original and adversarial dataset using Eq. 12.

The results are presented in Table 2. Overall, we find that both LLM judges and GloVE explanations are highly robust to adversarial attacks. We find that the attacks led to small drops in performances across datasets and judge models, with all performance drops below 0.1.

Table 3: Ablation study results

(a) GraniteGuardian3.2:5b

| | Perf. Degradation | | Fidelity (Acc.) | | Fidelity (F1) | |
|---|---|---|---|---|---|---|
| | GloVE | GloVE_NoFR | GloVE | GloVE_NoFR | GloVE | GloVE_NoFR |
| AgentHarm | -0.02 | **-0.70** | **0.88** | 0.05 | **0.93** | 0.0 |
| BeaverTails | **-0.09** | 0.17 | **0.69** | 0.64 | 0.67 | **0.78** |
| HarmBench | 0.07 | **0.0** | 0.93 | **1.0** | 1.0 | 1.0 |
| OpenAIMod | **-0.07** | -0.05 | **0.74** | 0.72 | **0.72** | 0.71 |
| SafeRLHF | **0.0** | 0.14 | **0.8** | 0.73 | **0.87** | 0.85 |
| SST | 0.0 | **-0.94** | 0.99 | 0.0 | 0.99 | 0.0 |
| XSTest | 0.0 | 0.0 | **0.71** | 0.62 | **0.80** | 0.59 |

(b) LlamaGuard3.3:8b

| | Perf. Degradation | | Fidelity (Acc.) | | Fidelity (F1) | |
|---|---|---|---|---|---|---|
| | GloVE | GloVE_NoFR | GloVE | GloVE_NoFR | GloVE | GloVE_NoFR |
| AgentHarm | -0.02 | **-0.06** | 0.93 | 0.93 | **0.97** | 0.96 |
| BeaverTails | 0.07 | **0.01** | **0.75** | 0.57 | **0.68** | 0.53 |
| HarmBench | -0.02 | **-0.05** | **0.96** | 0.95 | **0.98** | 0.97 |
| OpenAIMod | **-0.19** | 0.27 | 0.56 | **0.73** | **0.60** | 0.0 |
| SafeRLHF | **-0.05** | 0.54 | **0.77** | 0.32 | **0.80** | 0.08 |
| SST | -0.01 | -0.01 | 0.99 | 0.99 | 0.99 | 0.99 |
| XSTest | 0.0 | 0.0 | 0.57 | **0.69** | 0.63 | **0.69** |

**Ablation Studies.** We perform ablation studies to evaluate the effect of individual components on the performance of GloVE algorithm. Specifically, we create a baseline GloVE_NoFR by removing the verification step from the GloVE algorithm. These results are presented in Table 3. When explaining the GraniteGuardian, removing the FactReasoner component leads to decrease in fidelity across all datasets. When explaining LlamaGuard removing the FactReasoner component leads to significant decrease in fidelity on BeaverTails and SafeRLHF datasets, both in accuracy and F1 score. On AgentHarm, HarmBench and SimpleSafetyTests removing FactReasoner does not affect the fidelity or performance degradation metrics significantly. Finally, on OpenAIMod and XSTest test, removing FactReasoner actually leads to increase in fidelity accuracy. However, at the same time GloVE_NoFR achieves significantly lower F1 fidelity score on OpenAIMod.

**User Study.** We conduct a user study to compare explanations generated by GloVE and the baseline approach GELPE of the LLM-as-a-Judge behavior on the XSTest dataset. We recruited 18 participants internally from our organization and split them into two groups, with one group receiving explanations generated by GELPE and the other generated by GloVE. During the study users are shown 10 prompts and asked to predict whether the LLM-as-a-Judge would classify these as harmful or harmless. We measure participants' accuracy in predicting LLM-as-a-Judge decisions as a proxy for their understanding of its policy. Finally, users are asked to report their satisfaction by rating properties of the explanation satisfaction metrics Hoffman et al. (2023) on a 1-5 Likert scale.

Participants that have seen GloVE explanations achieved accuracy of $60\%$, compared to $58\%$ achieved by participants in the baseline condition. We conducted a non-parametric one-tailed Mann-Whitney test and find no significant difference between the two conditions. We find that GloVE led to higher user satisfaction on all metrics, and after conducting a Mann-Whitney statistical testing we find slight statistical significance in perceived usefulness ($p = 0.04$). Detailed user study description and additional results are included in Appendix.

## 7 CONCLUSIONS

In this work, we address the problem of generating high-level, verifiable global explanations for LLM-as-a-Judge. We designed an explanation pipeline consisting of two algorithms – CLoVE and GloVE, which summarize LLM-as-a-Judge policy while providing per-instance verification. We evaluated the explanations generated by the pipeline on a harm detection task, and find it produces concise rule sets that outperformed the baseline approach on quantitative metrics. Additionally, through a user study we found marginal improvement in users' understanding of LLM-as-a-Judge policy, and significant improvement in perceived usefulness compared to the baseline approach.

## 8 LIMITATIONS

One limitation of our work is that the quality of concepts and explanations generated by CLoVE and GloVE is limited by the quality of individual components in the pipeline. While we address the potential issue of hallucinations of the LLM-based generator and labeler components via verification, we acknowledge there are additional issues such as bias or prompt sensitivity that these components can inherit from their implementation. Additionally, we find that while GloVE generate highly faithful explanations, the participants struggled to comprehend them, highlighting the tradeoff between faithful and interpretable global explanations. In future work, we hope to investigate how GloVE explanations can be simplified to enable user understanding while maintaining high fidelity.

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
