# OpenReview forum: "Interpreting LLM-as-a-Judge Policies via Verifiable Global Explanations"
_ICLR.cc/2026/Conference — Submitted to ICLR 2026_

### Official Review · Reviewer_rX85 · 2025-10-31

**Soundness:** 3
**Presentation:** 3
**Contribution:** 2
**Rating:** 6
**Confidence:** 2

**Summary:**

This paper addresses the opacity and lack of interpretability in LLM-as-a-Judge systems by proposing a two-stage pipeline to extract verifiable global decision policies. It proposes a local explanation method that generates contrastive rationales in a BECAUSE-DESPITE format, and a  global algorithm that aggregates CLoVE’s local explanations into a compact policy.  The authors evaluate their method on 7 harm detection datasets (e.g., BeaverTails, HarmBench) using two LLM judges (GraniteGuardian3.2, LlamaGuard3.3), showing high fidelity to the judges’ decisions (F1 > 0.7 on most datasets) and robustness to paraphrasing/adversarial attacks. A small user study finds marginal improvements in policy understanding and significant gains in perceived usefulness over the GELPE baseline.

**Strengths:**

- The combination of contrastive reasoning (BECAUSE–DESPITE) and verifiable concept extraction is original. The integration of CLoVE and GloVE offers a systematic pipeline for producing both local and global interpretability, which goes beyond existing rule-based or concept-bottleneck approaches.

- GloVE’s use of FactReasoner to ensure that summarized concepts are probabilistically entailed by local ones adds an important layer of rigor. The inclusion of formal lemmas on graph homomorphism and entailment fidelity contributes to theoretical depth and credibility.

**Weaknesses:**

- Both CLoVE and GloVE rely on multiple LLM submodules (generator, verifier, labeler). This introduces possible circular dependencies, bias propagation, and sensitivity to prompt design, which are only briefly acknowledged in the limitations section.

- The evaluation focuses exclusively on harm detection. Although this is a relevant benchmark, the method’s generalizability to other LLM-as-a-Judge domains (e.g., summarization quality, factual accuracy evaluation) remains untested and should be explored in future work.

**Questions:**

n/a

---

> ### Author Response · Authors · 2025-11-22
>
> > Re: multiple LLM components / focus on harm detection
>
> We thank the reviewer for their thoughtful review and we appreciate their recognition of IF-DESPITE rules as original and impactful. Regarded the stated weaknesses,  we appreciate these points and agree they deserve further discussion.
>
> First, using several LLM-based components (generator, verifier, labeller, FactReasoner) does indeed introduce opportunities for bias propagation. Our design choice is to decompose the task into narrow, verifiable subtasks, and then to check the outputs of these components whenever possible (via alignment with important tokens and entailment). This does not eliminate the risk of bias or prompt sensitivity, but we feel that it helps to constrain it. The rules expose the concepts the judge relies on and the trade-offs it makes, which can then be audited, critiqued, or compared across judges. We will expand the limitations section to be more explicit about these residual risks and the kinds of failures that might still arise.
>
> Second, our empirical evaluation focuses on harm detection because this is a safety-critical domain with multiple public benchmarks and because guardrail models remain imperfect on many harm dimensions. However, the pipeline is not specific to harm: it can, in principle, be applied to any LLMaaJ task where decisions can be phrased in terms of discrete labels and concept-level rationales (e.g., answer relevance, factuality, social bias, profanity, or stylistic quality). We will clarify this intended generality in the main text and add a short discussion of how the method could be instantiated in other domains (acknowledging that demonstrating such generalisation empirically is an important direction for future work).

---

### Official Review · Reviewer_kaTn · 2025-10-31

**Soundness:** 2
**Presentation:** 2
**Contribution:** 2
**Rating:** 2
**Confidence:** 4

**Summary:**

This study presents an LLM-based method for extracting local and potentially global rules from LLM-as-a-Judge models to interpret their opaque decision-making processes. The approach comprises three components: a concept generator, a word-level explanation module, and a verifier, each of which relies on either an LLM or other specialized NLP models. Experiments conducted across multiple datasets demonstrate the effectiveness of the proposed method in extracting rules with high fidelity and accuracy, outperforming several baseline approaches. In my view, the main contribution of this work lies in its introduction of a pipeline for empirically deriving rules from LLM-as-a-Judge models to explain their decisions. This is particularly valuable given the growing use of such models for automated evaluation in various scenarios that would otherwise require substantial human effort.

**Strengths:**

1. The study introduces a method that generates logical, rule-based explanations to elucidate the often opaque decision-making process of LLM-as-a-Judge models. This approach is valuable for developing more formal and rigorous explanations of LLM behaviors.

2. The proposed method is presented clearly, making the overall framework easy to follow.

**Weaknesses:**

1. The supplementary materials appear to be incomplete. Specifically, the document containing the proofs for the lemma introduced in the study was not provided.
2. The application of global rules is not clearly explained, particularly in relation to the non-trivial graph-building process developed in the method.
3. The description of the dataset usage is unclear. It is not specified which portion of the data was used for rule extraction and which portion was reserved for testing.
4. The adversarial attacks introduced in the study are relatively trivial. It is unsurprising that minor word-level manipulations significantly degrade detection performance, and the greater impact of paragraph-level rephrasing—which alters more words—further supports this expectation. As a result, these attacks do not adequately justify the method's robustness.
5. The ablation study indicates that the verification step does not consistently improve performance. This raises questions about its necessity and practical contribution, which should be more explicitly discussed to clarify the key takeaways regarding its importance.

**Questions:**

My questions have been stated in the detailed review.

---

> ### Author Response · Authors · 2025-11-22
>
> We thank the reviewer for the helpful review.
>
> > Re: incomplete supp material
>
> We thank the reviewer for pointing out the missing material. We will update the supplementary material to include these proofs, and add explicit references from the main text to the corresponding sections in the appendix.
>
>
> > Re: application of global rules / graph-building process
>
> We are not entirely sure what this comment is referring to. If it is an issue with clarity, we can add a small, running example that starts from two or three CLoVE explanations, shows how they are represented as nodes and edges in the K-partite graph (Fig. 2), then illustrate how clustering/merging leads to a higher-level concept. We can additionally (explicitly) walk through how a single node in the final graph corresponds to an “IF u DESPITE v_1,..., v_L” rule, and how concepts u and v_j in a new input is checked at inference time.
>
> We are happy to discuss further if the reviewer can provide more information on what was the source of confusion.
>
> > Re: dataset usage
>
> In our current experiments, we use the same dataset split both to derive the rules and to evaluate fidelity. This is intentional since the rules are not intended to replace the LLMaaJ or to generalise beyond the domain that they are extracted from. The goal is to obtain a high-level, interpretable characterisation of the judge’s behaviour on a specific dataset of interest.
>
> For this use case, the primary requirement is that the rules faithfully capture the judge’s decisions on that dataset, rather than out-of-distribution generalisation. Evaluating fidelity on the same split from which the rules are derived is therefore aligned with the intended purpose of the method. We will clarify this design choice explicitly in the experimental setup section and, where possible, report additional experiments that apply the extracted rules to related but distinct datasets to illustrate how they behave out of distribution.
>
>
> > Re: trivial attacks
>
> We disagree with the reviewer that minor word-level manipulations "significantly degrade detection performance". As shown in Table 2, simple word-level modifications such as removing spaces, changing case, inserting punctuation or swapping words lead to very small drops in accuracy for both the LLMaaJ and GloVE (all below 0.1 across datasets and judges). The paraphrasing strategies (HIDE, ELABORATE, SUBSTITUTE) have a much stronger impact, esp on the judge, which is what one would expect.
>
> Our intention is to evaluate whether the rules inherit, or possibly improve upon, the robustness of the underlying judge under various perturbations (not to claim robustness against adversaries). The results show that: (i) for simple word-level noise, GloVE and the judge behave similarly, and (ii) under paraphrasing, GloVE’s accuracy drops less than the judge’s (e.g., a drop of 0.10 for GloVE vs up to 0.29 for the judge on OpenAIMod). We will clarify in the paper that this is partial evidence about robustness, but that analysis under stronger settimgs is an important direction for future work.
>
> > Re: effect of verification / ablation
>
> We agree that the impact of verification should be discussed more explicitly. As shown in Table 3, removing the FactReasoner component (GloVE_NoFR) typically reduces fidelity, sometimes dramatically. For example, for LlamaGuard on SafeRLHF, it drops from 0.80 to 0.08.
>
> There are a few cases where the accuracy of GloVE_NoFR is slightly higher (for OpenAIMod and XSTest), but this comes at the cost of much worse F1. We interpret this as evidence that verification via FactReasoner is important for high-fidelity rules, even if it may have a mixed effect on accuracy in some settings. We will extend the discussion in the paper to make this trade-off clearer.

---

### Official Review · Reviewer_prPU · 2025-10-31

**Soundness:** 3
**Presentation:** 4
**Contribution:** 3
**Rating:** 4
**Confidence:** 4

**Summary:**

The paper addresses the critical problem of opacity in "LLM-as-a-Judge" systems, whose black-box nature raises concerns about bias, reliability, and fairness. The authors argue that existing local explanations (like Chain-of-Thought) are often unreliable and fail to provide a high-level view of the judge's overall policy. They propose a two-part pipeline and evaluate it on seven harm-detection datasets, using LlamaGuard and GraniteGuardian as judges. They find the extracted GloVE policies are highly faithful to the original judge's decisions and robust to various text perturbations.

**Strengths:**

1. The "BECAUSE... DESPITE..." contrastive format is a significant strength. It moves beyond simple, one-sided rationales and captures the inherent ambiguity of moderation tasks, which is crucial for understanding why a judge made a specific trade-off.
2. The pipeline's core novelty is its focus on "verifiable" explanations. It doesn't just trust a generated explanation, it actively verifies it. CLOVE verifies concepts against important input words (via LIME), and GloVE verifies its clustered concepts (via FactReasoner). The ablation study (Table 3) successfully demonstrates that this verification step is crucial for maintaining fidelity.
3. The paper introduces clear, quantitative metrics for success, "Fidelity" and "Performance Degradation", to measure how well the extracted policy mimics the original judge. The GloVE pipeline shows very high fidelity (Table 1), demonstrating that the summarized policy is a very accurate "distillation" of the original LLM-judge.
4. The authors go beyond simple fidelity checks to test the robustness of the GloVE-extracted rules against text paraphrasing and adversarial attacks. The finding that the GloVE rules are often more robust than the original LLM-judge is an interesting result.

**Weaknesses:**

1. The primary weakness is that the proposed solution to LLM opacity is, itself, a highly complex and opaque pipeline of multiple LLMs. To explain one judge ($M$), the CLOVE algorithm uses an LLM generator ($G$), an LLM verifier ($V$), and the GloVE algorithm uses an LLM labeler and an LLM-based FactReasoner. This doesn't solve the opacity problem, it just shifts it. How do we know the verifier's policy or the FactReasoner's policy is unbiased? The final explanation is now dependent on the "judgment" of several other black-box models.
2. The ultimate goal of explainability is to improve human understanding. The paper's user study fails to show this. Participants' accuracy in predicting the judge's behavior was "marginal" and not statistically significant (60% for GloVE vs. 58% for the baseline). This finding suggests that the "highly faithful" GloVE explanations do not actually help humans understand the judge's policy any better than the baseline. The paper highlights "perceived usefulness", but this is a much weaker claim than demonstrating improved comprehension.
3. The full pipeline appears to be exceptionally complex and computationally expensive. Generating a single global policy requires running the base judge ($M$), plus a generator ($G$), LIME ($L$), and a verifier ($V$) for every single local explanation, followed by iterative clustering, labeling, and FactReasoning for the global policy. The paper provides no analysis of this cost, which seems prohibitive for practical use.
4. The CLOVE algorithm's "verifiability" is critically dependent on a local word-based explainer like LIME. LIME and similar feature-attribution methods are known to be unstable themselves. The paper does not analyze how sensitive the final global policy is to the instability or failure of this underlying component.

**Questions:**

1. The pipeline uses several LLMs (G, V, FactReasoner) to explain the target LLM ($M$). How can we trust the explanation if the components of the explainer are themselves opaque LLM-judges? Does this not simply defer the problem of bias and reliability to the explanation pipeline itself?
2. The user study showed no significant improvement in human understanding (60% vs 58% accuracy). Given that the primary motivation of the paper is to make opaque policies "transparent and interpretable", how do the authors interpret this negative result?
3. Could the authors provide an analysis of the computational cost (e.g., total LLM calls or time) required to generate one global policy? How does this cost scale with the number of local explanations?
4. How robust is the CLOVE method to the choice of the local word-based explainer ($L$)? What happens if LIME provides incorrect or unstable word attributions, and how does that error propagate to the final global policy?

---

> ### Author Response · Authors · 2025-11-22
>
> We appreciate the reviewer’s thoughtful and constructive comments. Below, we provide detailed responses to each point raised.
>
> > Re: opaque components
>
> Our aim is not to eliminate opaqueness fully but to replace a single judge with a structured and verifiable surrogate w/ a (more) explicit decision process.
>
> To clarify our approach for mitigating the risks associated with using LLMs:
> - We restrict each component to a specific task (e.g., concept proposal, concept verification, cluster labelling), instead of asking a single model to do everything at once (breaking things down is well-known to improve reliability for complex tasks [4, 5])
> - We run these components through verification (at the local level, we require that each concept must be supported by high-importance tokens from the judge; at the global level, candidate cluster labels must be entailed with high probability from FactReasoner before being accepted).
> - We evaluate the rules through fidelity and robustness metrics (and remove/ablate verification) and show that the verification step substantially improves fidelity on most datasets (Table 3).
>
> We fully agree that this approach does not completely remove bias or opacity in the LLMs (we already highlight this limitation in the paper). Our claim is more modest, that the proposed pipeline yields a rule-based approx. of the judge’s policy whose behaviour can be inspected in ways that are not possible for the original LLMaaJ.
>
> > Re: interpretation of the user-study
>
> We agree that this is an important point and will clarify this in the paper.
>
> Global surrogate explanations for complex black-box models are inherently hard for users to parse due to having to encode a complicated decision policy into a finite set of rules. We do not expect users to internalise the entire policy and simulate the judge perfectly. The small increase in predictive accuracy in our study is not surprising. Despite this, the study still demonstrates a positive effect on comprehension, suggesting that the distilled rules are valuable even in their current form.
>
> We view this work as a first step towards effective global explainability for LLMaaJ. In future work we can investigate other rule representations, visualisations, and interactive tools but this brings us pretty far outside the scope of our current contribution.
>
> > Re: complexity and computational cost
>
> We agree that the cost analysis should be explicit. However, the pipeline follows a standard pattern of global explanation methods. GELPE [1], GLocalX [2], and Lloom [3] extract local explanations for individual inputs and then summarise them into a global surrogate.
>
> In terms of LLM calls, generating a local explanation for a single input requires one call to the judge, two calls to the Generator (1/class), and at most six calls to the Verifier (up to 3 concepts/class), resulting in at most eight LLM calls per instance. The global stage clusters the resulting concepts; for each cluster we call the labeller up to 10 times to propose candidate labels, and use FactReasoner to score entailment. This global summarisation is run once per dataset (not per query).
>
> We want to emphasise that this pipeline is not intended to be used in an online faashion. It is more useful to think of the pipeline for offline analysis of an LLMaaJ on datasets (e.g.., borderline or safety-critical cases), where the computation reqs are manageable. We can report approximate times and total LLM calls for our experiments in the final version if desired.
>
> > Re: dependence on the local word-based explainer
>
> To answer how robust GloVE pipeline is to the choice of local explainer we ran the pipeline with SHAP and report the results here:
>
> Dataset | Performance degradation | Fidelity (Acc) | Fidelity (F1)
>
> BeaverTails | 0.11 | 0.72 | 0.64
>
> OpenAIMod | 0.02 | 0.71 | 0.78
>
> SafeRLHF | 0.03 | 0.84 | 0.86
>
> AgentHarm | 0.0 | 0.91 | 0.95
>
> SimpleSafetyTests  | -0.01 | 0.99 | 0.99
>
> HarmBench | -0.01 | 0.97 | 0.98
>
> XSTest | 0.0 | 0.75 | 0.79
>
> The results are based on explanations generated for GraniteGuardian model.
>
> These scores are very close to those reported in the paper with LIME (Table 1), wrt perf degradation and fidelity. This suggests that the global rules (GloVE) are robust to reasonable changes in the local explainer, and any instability of token-level attributions does not affect the quality of the distilled policy. We will include these SHAP-based results in the appendix.
>
> [1] Agiolo et al. From large language models to small logic programs: building global explanations from disagreeing local post-hoc explainers
> [2] Setzu et al. GLocalX - From Local to Global Explanations of Black Box AI Models
> [3] Lam et al. Concept induction: Analyzing unstructured text with high-level concepts using lloom
> [4] Zhang et al. When Splitting Makes Stronger: A Theoretical and Empirical Analysis of Divide-and-Conquer Prompting in LLMs
> [5] Wei et al. Chain-of-thought prompting elicits reasoning in large language models

---

### Official Review · Reviewer_ibRx · 2025-10-31

**Soundness:** 2
**Presentation:** 2
**Contribution:** 2
**Rating:** 4
**Confidence:** 3

**Summary:**

This paper attempts to provide an explanation of the thought process of LLM-as-a-judge in detecting harmful information by analyzing its potential decision-making process. The paper proposes a method for extracting high-level concepts from raw text, including CLoVE and GLoVE. CLoVE analyzes the positive and negative meanings of sentences to extract contrasting explanations, while GLoVE gradually aggregates local contrasting explanations to form global explanations. This helps explain the decision-making process of LLM-as-a-judge when handling complex texts, as it can be achieved through an understanding of the overall text.

The definition of the task in this paper is somewhat confusing. The proposed method evaluates performance by comparing the analysis results of harmful texts using this method with those of other harmful content detection models, and then uses this consistency as an explanation for the interpretability of other harmful content detection models. This highly indirect approach raises doubts about the rationality of the task and experimental design.

**Strengths:**

This paper attempts to incorporate more fine-grained analysis into the harmful detection process, including comparative explanations based on opposing perspectives and organizing local contexts into global textual explanations. This approach helps improve the performance of harmful detection and provides a certain degree of procedural transparency.

**Weaknesses:**

The task definition in this paper is confusing and poorly presented.

This paper focuses on explaining and analyzing the original text in harmful detection, rather than analyzing the output results of LLM-as-a-Judge, then exhibiting their potential bias. Therefore, I do not consider this to be explanatory work on LLM-as-a-Judge, but rather more like a standalone LLM harmful detection model.
It may not qualify as explanatory research on defense LLMs like LlamaGuard, as the experiments in this paper compare the analysis results of GLoVE with the analysis content of other LLM-as-a-Judge models. This is quite perplexing: why should your model's analysis of the text need to correlate with the analysis of any other model?

If the task of this paper, as mentioned in line 338—"Our experiments focus on explaining LLM-as-a-Judge on the task of content harm detection"—were placed earlier, such as in the introduction or abstract, it might be better. Otherwise, readers might assume you are studying the explanatory aspects of LLM-as-a-Judge for general text.

Based on my understanding of the task definition as mentioned above, the current baselines are entirely insufficient. At the very least, there should be a comparison with LlamaGuard's capability in harmful content detection.

Additionally, if my misunderstanding is due to my oversight, I apologize and would appreciate it if the authors could provide further clarification. For now, I will give a low confidence.

**Questions:**

- What backbones are the Generator, Verifier, and Local word-based explainer mentioned in this paper? Has the performance of these components been individually quantified and evaluated?

- How are important words defined? Has the accuracy of extracting these important words been evaluated?

---

> ### Author Response · Authors · 2025-11-22
>
> We thank the reviewer for their thoughtful review. We would like to clarify some points.
>
> > Re: task definition / what is being explained
>
> Regarding the task definition, the goal of GloVE is to analyse and explain the decisions of a fixed LLM-as-a-Judge (LLMaaJ), not to build a separate harmful-content detector for the text itself. The generator proposes high-level candidate concepts from the text, but these are only retained if they are supported by words that are important for the judge’s decision. The local word-based explainer (LIME in our experiments) identifies the tokens that most strongly influence predictions for each label. The verifier then filters out any concept that is not grounded in this set of important words. The final concepts in the BECAUSE/DESPITE rules are those that trigger harmful or harmless decisions for the LLMaaJ, rather than arbitrary harmful or harmless notions present in the text. The global rules produced by GloVE then summarise these concept-level explanations across many instances into an explicit approximation of the judge’s policy.
>
> > Re: baselines and task framing
>
> We believe that the reviewer's claim that the baselines are insufficient reflects a difference in task framing. Our objective is not to outperform LlamaGuard or GraniteGuardian as harm detectors, but to explain their decisions via a global, concept-based surrogate. In our experiments, these guardrail models play the role of fixed reference judges whose outputs we treat as ground truth for the purpose of explanation. GloVE and the baseline method GELPE are then evaluated purely in terms of how faithfully they reproduce the judges’ policies (performance degradation and fidelity), not in terms of absolute harm-detection accuracy.
>
> Within this framing, the relevant comparison is between different global explanation methods applied to the same LLMaaJ, rather than between different harm-detection models.
>
> > Re: backbones/component eval
>
> Both the Generator and Verifier are instantiated with llama-3.3-70b-instruct, and the local word-based explainer is LIME. We do not evaluate the Generator and Verifier as standalone models on separate benchmarks, because they are not used as independent predictors. Their quality is assessed indirectly through the performance of the full pipeline, i.e., through the fidelity and robustness of the resulting rules to the LLMaaJ’s decisions and through the ablation study that removes the verification step in GloVE (Table 3). We will clarify these steps.
>
> > Re: definition and eval of important words
>
> Important words are defined via LIME: for each input and label, we apply LIME to the LLM-as-a-Judge and take the top-ranked tokens according to their contribution to the model’s predicted probability for that label. These tokens form the set of important words used to verify candidate concepts in CLoVE.
>
> We do not evaluate the token attributions in isolation but within the pipeline. The high fidelity of the extracted rules to the judge’s decisions suggests that the important words capture the main evidence used by the judge. In our experiments (see response to Reviewer prPU's comment on the local word-based explainer) we replace LIME with SHAP as the local explainer and obtain very similar performance degradation and fidelity scores across all datasets. This indicates that the overall pipeline is robust to the specific choice of local word-based explainer, and that the notion of “important words” is not brittle to the underlying attribution method.

---

### Meta-Review · Area_Chair_iVAm · 2026-01-06

**Summary:**

This submission proposes an approach for extracting high-level concept-based global policies from LLM-as-a-Judge. Basically, the idea is interesting. However, the solution is somehow straightforward.

**Reviewer Concerns:**

1. Missing supplementary proofs / incomplete materials, this is raised by: kaTn. The Authors’ response: Will update supplementary materials to include proofs and add explicit references. But, no proofs are updated.

2. Limited domain: only harm detection; generalizability untested. This is raised by: rX85 that only harm detection; does it generalize to other LLM-as-judge domains.

The authors’ response: They argue pipeline is domain-agnostic in principle for label-based judging tasks; will clarify and add discussion; empirical generalization is future work.
If that is the thing, i think the title is over-claimed.

Basically, the rebuttal need significantly improved to enhance the submission.

**Reviewer Scores:**

Three negative (one with 2). Only one positive (6). In the rebuttal, no clear evidence the reviewers would raise score.

---

### Decision · Program_Chairs · 2026-01-26

Reject